# Investigation of the Effects of Nanoparticle Concentration and Cutting Parameters on Surface Roughness in MQL Hard Turning Using MoS$_2$ Nanofluid

**Ngo Minh Tuan** [1] , **Tran Bao Ngoc** [2], **Tran Le Thu** [3] **and Tran The Long** [1,*]

[1] Department of Manufacturing Engineering, Faculty of Mechanical Engineering,
Thai Nguyen University of Technology, Thai Nguyen 250000, Vietnam; minhtuanngo@tnut.edu.vn

[2] Department of Fluid Mechanics, Faculty of Automotive and Power Machinery Engineering,
Thai Nguyen University of Technology, Thai Nguyen 250000, Vietnam; baongoctran@tnut.edu.vn

[3] Library Information Technology Department, Thai Nguyen University of Technology,
Thai Nguyen 250000, Vietnam; tranlethu@tnut.edu.vn

* Correspondence: tranthelong90@gmail.com or tranthelong@tnut.edu.vn; Tel.: +84-985-288-777

**Abstract:** Minimum quantity lubrication (MQL) has gained significant attention in various research fields and industrial applications for its advantages of being environmentally friendly and suitable for sustainable production. The effectiveness of MQL is increasing significantly by using nano cutting fluid, which can be produced by suspending nanoparticles in the based cutting fluid. This study aims to investigate the effects of MoS$_2$ nanoparticle concentration, cutting speed, and feed rate on MQL hard turning of 90CrSi steel in terms of surface roughness and surface microstructure. The Box–Behnken experimental design was used to analyze the influence of input parameters and their interaction effects as well as to find the optimal set of variables. The obtained results prove the improvement of the machinability of carbide tools due to higher cooling and lubricating performance created by MoS$_2$ nanofluid MQL, which contributes to improve the surface quality and reduce the manufacturing cost. There is an interaction effect between nanoparticle concentration and feed rate which has a strong influence on surface roughness.

**Keywords:** hard turning; MQL; nanoparticles; nano cutting fluid; difficult-to-cut material; surface roughness



## 1. Introduction

In recent years, green and sustainable machining processes have gained growing concern in the metal cutting industry. Following this trend, the reduction or elimination of cutting oil usages will be the most effective way toward sustainable production; thus, the MQL method is a promising solution and is widely applied in machining processes [1]. In the MQL method, a very small amount of cutting oil is directly sprayed in oil mist form into the cutting zone to reduce the friction and improve the lubricating effect. Hence, it brings out a better surface quality and longer tool life than wet and dry machining methods [2–5]. The application of MQL in difficult-to-cut machining materials is still limited due to the low cooling performance. Some approaches have been studied and proven to improve this main drawback. In 2020, O. Pereira et al. [6] investigated milling performance using CO$_2$ cryogenic combined with MQL milling of Inconel 718. The authors concluded that, due to the better cooling effect, the cutting forces decreased by about 22% and tool life was prolonged by about 57% compared to MQL alone. The significant improvement in tool life when compared to CO$_2$ cryogenic and wet machining was reported in [7,8]. The effect of cutting heat was much smaller under the CryoMQL technique than that of the dry condition [9]. A. Rodríguez et al. [10] applied liquefied CO$_2$ for the drilling process of CFRP-Ti6Al4V. The authors concluded that the cutting temperature significantly decreased, hole surface quality improved, and tool life was much prolonged compared

with the dry condition. Another promising solution is that the use of nano cutting fluid rather than conventional ones to improve the cooling and lubricating effects; this has been considered as a novel solution to overcome the main problems of the MQL technique [11].

Nanofluids have been proven to have higher heat transfer ability as well as thermal and lubricating properties, so the machining performance is improved [11,12]. Sharma et al. analyzed and evaluated the effectiveness of the application of MQL with nano cutting oil in the machining process [13]. The obtained results showed that nano cutting fluid-based minimum quantity lubrication (NFMQL) method brought out better results in terms of the cutting temperature and surface roughness compared with the pure cutting oil-based MQL.

Among many types of nanoparticles, $MoS_2$ nanosheets are lightweight (about 4.8 g/cm$^3$) and have a high melting point (1185 $^\circ$C), low hardness (1–1.5 according to Mohs hardness scale), and excellent chemical and thermal stability. They also do not have any chemical reaction with the metal surface, even at high temperatures, so they are suitable for machining [14]. Recently, $MoS_2$ nanoparticles have been applied to suspend with the cutting oil to form $MoS_2$ nanofluid used in MQL for improving the cooling lubrication in machining [15], which has attracted the interest of many researchers. $MoS_2$ NFMQL helped to reduce the specific energy and surface roughness in grinding process [16]. P. Kalita et al. [17] studied and applied $MoS_2$ NFMQL to the grinding process of cast iron. The results also indicated the reduction of specific energy and improvement of surface quality compared to wet grinding. Z. Dongkun et al. [18] analyzed the effect on surface roughness and cutting force when applying three types of nanoparticles ($MoS_2$, $ZrO_2$, and carbon nanotubes (CNTs)). The analysis results showed that the cutting forces significantly reduced and surface topography profile values reduced about 10% when compared to dry grinding, in which $MoS_2$ NFMQL had the smallest specific grinding energy of $ZrO_2$ nanoparticles and carbon nanotubes due to the very low friction coefficient of $MoS_2$ nanosheets, which can provide a low coefficient of friction up to 0.03–0.05 or even lower [19]. Y. Wang et al. [19] also found that the $MoS_2$ nanosheets are ellipsoidal, and their layer structure is a hexagonal crystal system combining Mo and S through a covalent bond, which is short, but the spacing between sulfur atoms is large. The bond between two adjacent sulfur atom layers is weak; hence, "an easy-to-slide plane" will be generated from the weak binding of sulfur atoms between molecular layers by the shearing force caused by cutting processes. The numerous easy-to-slide planes contribute to reduce the friction coefficient in contact faces effectively. W.T. Huang et al. [20,21] investigated the grinding and micro milling processes under nanofluid MQL condition using multi-walled carbon nanotubes (MWCNTs) and $MoS_2$ nanosheets. The authors concluded that MWCNTs were suitable to reduce the grinding forces and temperature, while $MoS_2$ nanofluid was more favorable for improving surface quality due to its excellent surface activity and the fact that $MoS_2$ nanoparticles can quickly fill gaps to form a tribofilm and maintain the lubricating effect [22].

Y. Zhang et al. [23] studied the lubricating performance of $MoS_2$ NFMQL in grinding in terms of specific grinding energy and surface roughness. Compared with dry, flood, and pure MQL conditions, the specific grinding energy and surface topography profile values significantly decreased under the $MoS_2$ NFMQL condition. The study also investigated the different $MoS_2$ nano concentrations, including 1.0%, 2.0%, and 3.0%. The authors found that specific grinding energy and surface roughness initially increased and then decreased as $MoS_2$ nano concentration went up. A. Uysal et al. [24] studied the flow rate in MQL and $MoS_2$ NFMQL with vegetable oil as the based oil in the milling process of AISI 420 steel using uncoated carbide inserts. The results showed that the increase of flow rate brought out better results in terms of tool wear and surface roughness values. $MoS_2$ NFMQL revealed the higher cooling and lubricating effects compared to dry and pure MQL milling. B. Rahmati et al. [25] analyzed the efficiency of $MoS_2$ NFMQL in the end milling of aluminum alloy with different nano concentrations. The close relation between the concentration values of $MoS_2$ and the lubricating film formation was reported. The machined surface improved by using NFMQL due to the rolling, filling, and polishing

actions of MoS$_2$ nanoparticles at the tool–workpiece interface. In 2019, P.Q. Dong and his colleagues [26] evaluated the effectiveness of MoS$_2$ nano cutting oil used in the novel MQCL device based on the principle of a Ranque–Hilsch vortex tube in the hard milling of SKD 11 tool steel. The authors concluded that the MoS$_2$ nanoparticle concentration was the complicated function, which was very sensitive to the machined surface quality, and the formation of the MoS$_2$ tribofilm had a very close relationship with the concentration and interacted with cutting conditions.

However, the application of MoS$_2$ nanoparticles suspended in oil-in-water emulsion for MQL hard turning is still limited in terms of surface roughness, topography, and microstructure. Hence, the authors of the present work were motivated to analyze the influence of MoS$_2$ nanoparticle concentration, cutting speed, and feed rate on surface roughness in MQL hard turning of 90CrSi steel using coated carbide inserts. Experimental evaluations were conducted on the lubricating property of MoS$_2$ nanoparticle in terms of different concentrations and the interaction effects with cutting speed and feed rate on the machined surface.

## 2. Materials and Methods

*Experimental Set Up*

The experimental setup and diagram for this research are shown in Figures 1 and 2. The experiments were conducted on a CS-460 × 1000 Chu Shing lathe (Pin Shin Machinery Co., LTD, Taichung City, Taiwan). The MQL nozzle was arranged to spray on the flank face. An air pressure regulator and air flow rate valves were used to stabilize and control the pressure and air flow. When the high-pressure air flow passes through the MQL nozzle, it will create a vacuum area, and based on the capillary principle, the nano-cutting oil is sucked up. In there, the nano cutting oil is shredded into small droplets in the form of oil mist by a high-pressure air flow (Figure 2).

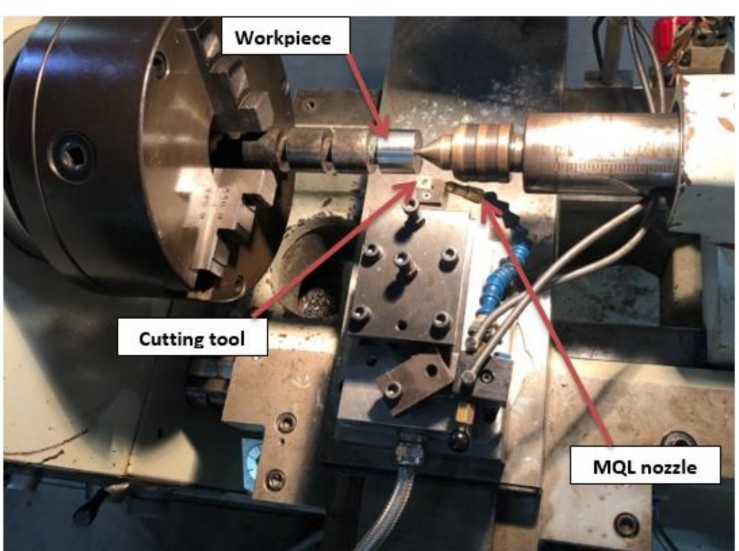

**Figure 1.** Experimental setup.

90CrSi low alloy steel is commonly used to manufacture machine parts requiring high strength, wear resistance, and tensile or subjected to high impact load. The 90CrSi hardened steel samples were obtained from North of Jinchuanyuan, Changzhuang Town, Tangshan, China with a hardness of 60–62 HRC, a diameter of 40 mm, and a length of 250 mm; the chemical composition is shown in Table 1, while the physical and mechanical properties of 90CrSi steel are shown in Table 2. Tungaloy CCD coated carbide inserts (CNMG120404 TM T9125) were used. The cutting oil was oil-in-water emulsion oil (Em). MoS$_2$ nanoparticles manufactured by China Luoyang had a layered structure with an average size of 30 nm (the grain size was based on previous studies [25,27]) and a purity of >99.9%,

and their TEM image is shown in Figure 3. The nanoparticles were directly mixed with the base oil and ultrasonication with an ultrasonic frequency 40 kHz for 30–45 minutes [18,28] was performed by using an Ultrasons-HD Ultrasonicator to ensure the uniform distribution of $MoS_2$ nanoparticles. The obtained nano cutting fluid was then directly used for the MQL system [13,27]. For the MQL system, a NOGA Minicool MC1700 was used, and the MQL parameters were the air pressure, $p$ = 6 bar, and an airflow rate of 200 L/min [29]. The nozzle distance was 20 mm and the spray angle was 12° [30]. The cutting depth t = 0.15 mm was fixed. The surface roughness values of the machined surface were measured by a Surftest SJ-210 (Mitutoyo, Japan). A KEYENCE VHX-7000 Digital Microscope (KEYENCE Corporation, Osaka, Japan) was used to investigate the surface microstructure.

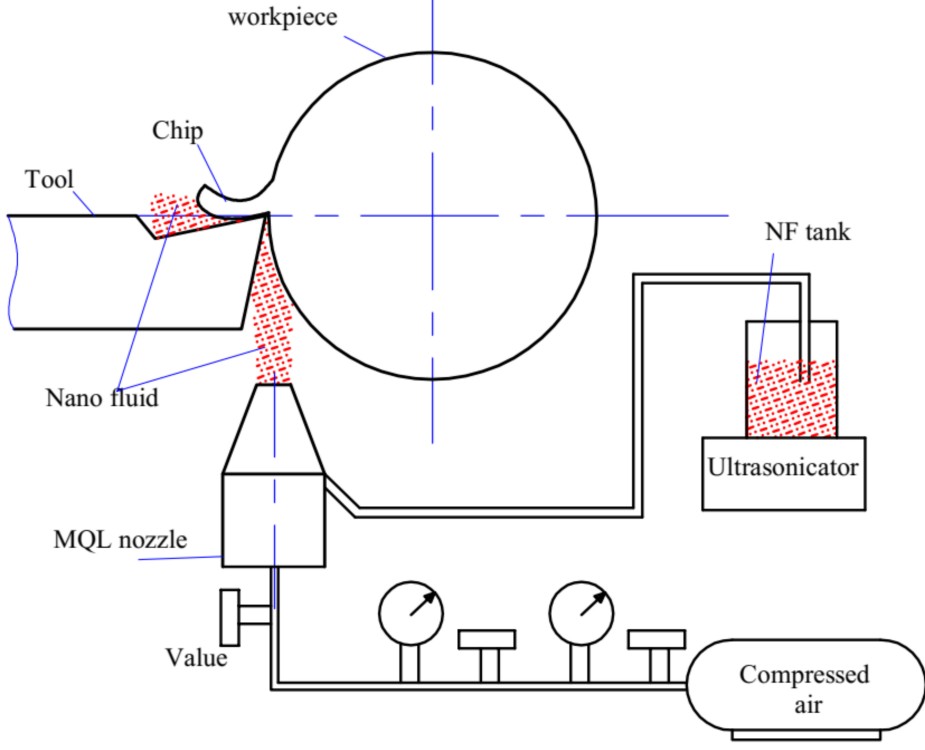

**Figure 2.** Experimental diagram.

**Table 1.** Chemical composition of 90CrSi steel.

| Element | C | Si | Mn | Ni | S | P | Cr | Mo | W | V | Ti | Cu |
|---|---|---|---|---|---|---|---|---|---|---|---|---|
| **Weight (%)** | 0.85 ÷ 0.95 | 1.20 ÷ 1.60 | 0.30 ÷ 0.60 | Max 0.40 | Max 0.03 | Max 0.03 | 0.95 ÷ 1.25 | Max 0.20 | Max 0.20 | Max 0.15 | Max 0.03 | Max 0.3 |

**Table 2.** Physical and mechanical properties of 90CrSi steel.

| Tensile Strength (MPa) | Yield Stress (MPa) | Young's Modulus (MPa) |
|---|---|---|
| 790 | 445 | 1.9 |

Box–Behnken designs are applied to generate higher order response surfaces with fewer runs than a normal factorial technique. This design is still considered to be more proficient and powerful than other ones, such as the three-level full factorial design. Hence, a Box–Behnken optimal experimental design with three input variables was used to build an experimental planning table to investigate the effects of nanoparticle concentration (NC), cutting speed (V), and feed rate (f) on surface roughness $R_a$. The values of the survey variables were based on previous studies [18,28] and are shown in Table 3.

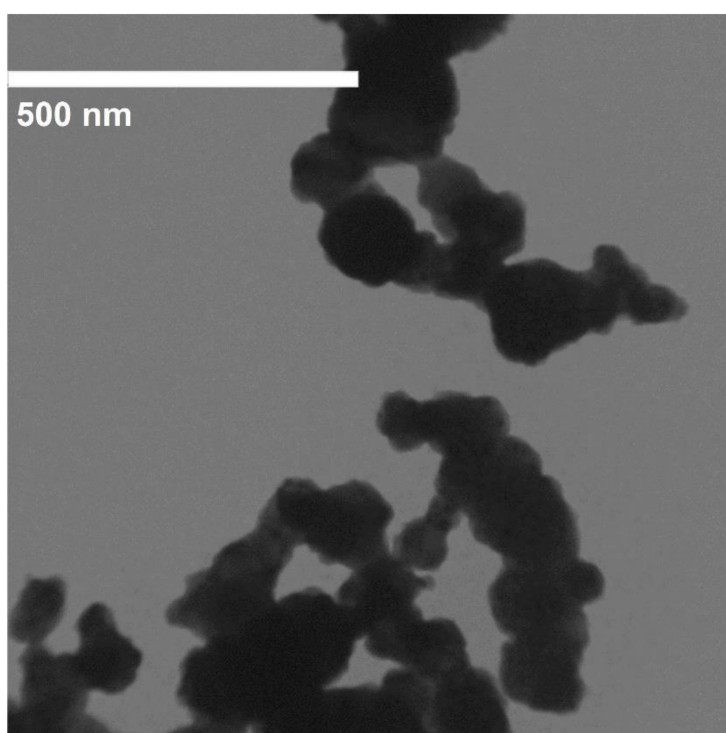

**Figure 3.** TEM image of MoS$_2$ nanoparticles manufactured by China Luoyang. Reproduced from [28], published by MDPI 2019.

**Table 3.** Input machining parameters and their three levels.

| Input Machining Variables | Symbol | Level | | | Response Variable |
|---|---|---|---|---|---|
| | | Low | Medium | High | |
| Nanoparticle concentration (NC), wt% | A | 1.0 | 2.0 | 3.0 | Surface roughness R$_a$ |
| Cutting speed (V), m/min | B | 80 | 120 | 160 | |
| Feed rate (f), mm/rev | C | 0.1 | 0.15 | 0.2 | |

## 3. Results and Discussions

The experimental matrix including 30 trials with three input variables was built by using a Box–Behnken design with the help of Design-Expert 11 software. All trials were carried out according to RunOrder in Table 4 and were repeated three times to take the average values. The surface roughness Ra was measured three times and taken by the average value after each trial by the Mitutoyo SJ-210 and the obtained results are shown in Table 4.

The ANOVA results for Ra with significance level $\alpha = 0.05$ were determined by using Design-Expert 11 software (Table 5). It can be clearly observed that the large Fisher coefficient (F-Value) for the survey model was 15.65 and the probability $p$ value was less than 0.05, which proves that the selected quadratic model is suitable, and there is only a 0.01% chance for this model to be affected by noise factors [31]. Feed rate (f), the interaction between nanoparticle concentration and feed rate (NC∗f), the interaction between cutting speed and feed rate (V∗f), and second-order interactions NC$^2$, V$^2$, and f$^2$ have the largest Fisher coefficients and their probability $p$ values were less than 0.05, showing the statistical significance and considerable influence on the objective function Ra. Variables (NC and NC∗V) with probability $p$ values greater than 0.1 had less influence on the predictive model. The Fisher F-value calculated for lack of fit was relatively small at 1.14, indicating that the predictive model is suitable and less affected by noise factors.

**Table 4.** Input parameters and experimental results.

| Std | Run | NC (wt%) | V (m/min) | f (mm/rev) | $R_a$ (µm) |
|-----|-----|----------|-----------|------------|------------|
| 1 | 27 | 1 | 80 | 0.15 | 0.766 |
| 2 | 14 | 3 | 80 | 0.15 | 0.730 |
| 3 | 5 | 1 | 160 | 0.15 | 0.600 |
| 4 | 9 | 3 | 160 | 0.15 | 0.686 |
| 5 | 15 | 1 | 120 | 0.1 | 0.504 |
| 6 | 26 | 3 | 120 | 0.1 | 1.157 |
| 7 | 24 | 1 | 120 | 0.2 | 1.767 |
| 8 | 18 | 3 | 120 | 0.2 | 1.111 |
| 9 | 23 | 2 | 80 | 0.1 | 0.614 |
| 10 | 3 | 2 | 160 | 0.1 | 0.432 |
| 11 | 29 | 2 | 80 | 0.2 | 0.948 |
| 12 | 4 | 2 | 160 | 0.2 | 1.397 |
| 13 | 16 | 2 | 120 | 0.15 | 0.805 |
| 14 | 8 | 2 | 120 | 0.15 | 0.912 |
| 15 | 19 | 2 | 120 | 0.15 | 0.452 |
| 16 | 6 | 1 | 80 | 0.15 | 0.843 |
| 17 | 1 | 3 | 80 | 0.15 | 0.858 |
| 18 | 12 | 1 | 160 | 0.15 | 0.573 |
| 19 | 2 | 3 | 160 | 0.15 | 0.654 |
| 20 | 28 | 1 | 120 | 0.1 | 0.548 |
| 21 | 22 | 3 | 120 | 0.1 | 1.050 |
| 22 | 10 | 1 | 120 | 0.2 | 1.599 |
| 23 | 21 | 3 | 120 | 0.2 | 1.432 |
| 24 | 13 | 2 | 80 | 0.1 | 0.589 |
| 25 | 25 | 2 | 160 | 0.1 | 0.381 |
| 26 | 17 | 2 | 80 | 0.2 | 0.820 |
| 27 | 30 | 2 | 160 | 0.2 | 1.037 |
| 28 | 20 | 2 | 120 | 0.15 | 0.535 |
| 29 | 7 | 2 | 120 | 0.15 | 0.939 |
| 30 | 11 | 2 | 120 | 0.15 | 0.821 |

A regression model to predict surface roughness was built and is shown in Equation (1). The relevance of the regression model to determine the surface roughness value was evaluated through the coefficient of determination $R^2$ (Table 6). The results show that the coefficient of determination $R^2 = 87.57\%$ and the adjusted coefficient of determination $R^2 = 81.97\%$ were fairly significant, proving that the regression model is consistent with the experimental data set. Predicted $R^2$ was not significantly different from the adjusted coefficient of determination Adj $R^2 = 74.70\%$ and the accuracy check coefficient (Adeq Precision) = 15.796, proving that the prediction model is appropriate and can be used to predict surface roughness values [31].

$$Ra = 0.67575 + 0.024125NC + 0.0182125V - 19.915f + 0.0005875NC*V - 4.945NC*f + 0.066V*f + 0.16925 + NC^2 - 0.0001246875V^2 + 93.1f^2 \quad (1)$$

**Table 5.** ANOVA analysis results.

| Source | Sum of | df | Mean | F-Value | *p*-Value | |
|---|---|---|---|---|---|---|
| Model | 3.078313 | 9 | 0.342035 | 15.65286 | <<0.0001 | Significant |
| NC | 0.01428 | 1 | 0.01428 | 0.653521 | 0.428369 | |
| V | 0.010404 | 1 | 0.010404 | 0.476128 | 0.498111 | |
| f | 1.461681 | 1 | 1.461681 | 66.89229 | <<0.0001 | |
| NC∗V | 0.004418 | 1 | 0.004418 | 0.202185 | 0.657798 | |
| NC∗f | 0.489061 | 1 | 0.489061 | 22.38134 | 0.000128 | |
| V∗f | 0.139392 | 1 | 0.139392 | 6.379128 | 0.020098 | |
| $NC^2$ | 0.211536 | 1 | 0.211536 | 9.680743 | 0.005501 | |
| $V^2$ | 0.29391 | 1 | 0.29391 | 13.45046 | 0.001529 | |
| $f^2$ | 0.400044 | 1 | 0.400044 | 18.30757 | 0.000367 | |
| Residual | 0.437025 | 20 | 0.021851 | | | |
| Lack of Fit | 0.073218 | 3 | 0.024406 | 1.140449 | 0.361035 | Not significant |
| Pure Error | 0.363807 | 17 | 0.0214 | | | |
| Cor Total | 3.515338 | 29 | | | | |

**Table 6.** Fit statistics.

| Fit Statistics | Value |
|---|---|
| $R^2$ | 0.87568 |
| Adjusted $R^2$ | 0.819737 |
| Predicted $R^2$ | 0.747006 |
| Adeq Precision | 15.7962 |

### 3.1. Independent Influence of the Input Variables on Surface Roughness Ra

Figure 4 shows the independent influence of the investigated variables on surface roughness $R_a$. The effects of nanoparticle concentration, cutting speed, and feed rate on surface roughness values all have inflection points, which indicate that the studied ranges are appropriate.

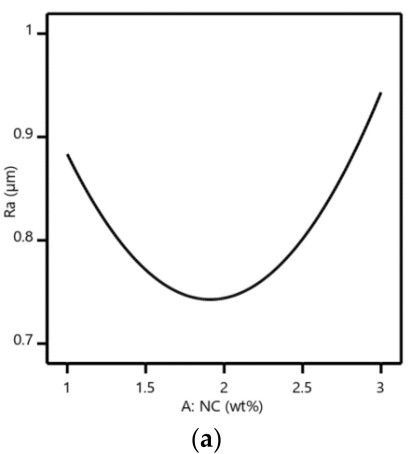

(**a**)

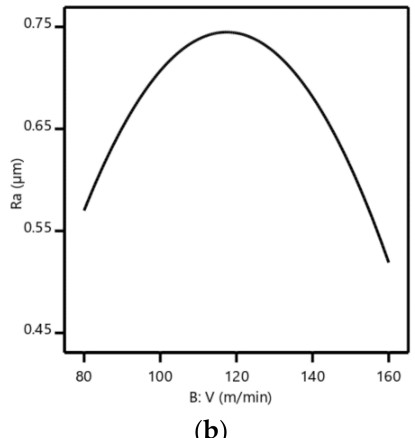

(**b**)

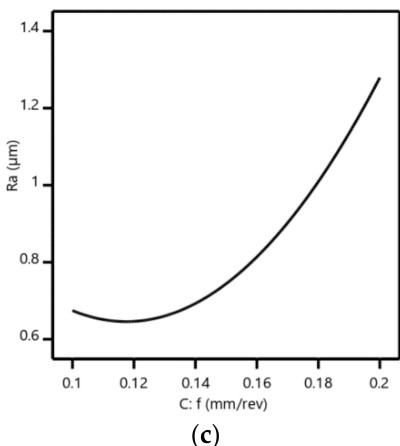

(**c**)

**Figure 4.** Effect of investigated factors on surface roughness: (**a**) nanoparticle concentration, (**b**) cutting speed, and (**c**) feed rate.

In Figure 4a, it can be seen that the surface roughness went up with increasing nanoparticle concentration and reached the minimum at the medium level of nanoparticle concentration (2.0 wt%), which agrees with the obtained results of Zhang Dongkun et al. [18]. There is an interaction effect of nanoparticle type and based cutting oil, so there exists an appropriate value of concentration for each type of nano cutting fluid to promote the best cooling and lubricating effect. For $MoS_2$ nanoparticles, the formation of tribofilm is very sensitive to the nano concentration value [26], so the increase in $MoS_2$ nanoparticle concentration from 2.0% to 3.0% reduced the distance among particles, and collisions among nanoparticles increased. Hence, the formation of tribofilm is unfavorable, which causes a negative effect on the lubricating performance. The nanoparticles also accumulate on the cutting edge, causing scratches on the machined surface [25,26,28]. On the other hand, turning is a machining process with an open cutting area and a large dynamic clearance angle, so nano cutting oil is dissipated into the surrounding environment and along the rake face, facilitating the chip to run efficiently. $MoS_2$ nanoparticles have a very good lubricating property, ellipsoidal morphology, and large surface area, so with a reasonable concentration, the nanoparticle easily adheres to the surface, creating favorable conditions for forming $MoS_2$ tribofilm in cutting zones [25,26]. This contributes to reducing the friction and tool scratching, thereby improving the surface roughness.

In Figure 4b, it can be clearly observed that cutting speed also influenced surface roughness. As the cutting speed increased from 80 to 120 m/min, the surface roughness increased, but it decreased when the cutting speed rose from 120 to 160 m/min; the reasons for this were previously discussed by [32,33]. At low cutting speeds, the heat generation required to reduce the shear strength of a workpiece material is not enough, so higher cutting forces will be required, affecting the surface finish negatively. In addition, the nanoparticles easily adhere to the top of the cutting edge, causing scratches on the machined surface. This phenomenon is formed and lost continuously, thus deteriorating surface quality [34]. For high cutting speeds, the workpiece shear strength decreases because the cutting heat is high enough to reduce the required cutting forces and positively affect the machined surface finish. On the other hand, nano cutting oil penetrates the cutting zone without being impeded and does not adhere to the top of the cutting edge, so the surface roughness value decreases. The effect of feed rate on surface roughness is the strongest; as feed rate per tooth increases, the distances between peaks and valleys will be higher [1,29], and in hard machining, the formation of surface roughness is mainly due to the scratches of the cutting tool on the machined surface, and the influence of other factors is not significant. Surface roughness rapidly went up with the increase of feed rate (Figure 4c). However, the investigation of the feed rate in the survey model was necessary to determine the interaction effect between the feed rate and other factors, and at the same time to analyze the influence of these factors in the roughing or finishing conditions.

### 3.2. Interaction Effects between Input Machining Variables on the Surface Roughness

The interaction effect between the investigated variables on the surface roughness value is shown in Figure 5. Figure 5a shows that the interaction between nanoparticle concentration and cutting speed had little effect on surface roughness, which reached the minimum value with a nanoparticle concentration of 2.0 wt% even when changing the cutting speed [18]. Figure 5b depicts the interactive effect between nanoparticle concentration and feed rate. Surface roughness reached the minimum value at low levels of nano concentration and feed rate. The surface roughness value was proportional to NC with a low feed rate (0.1 mm/rev) but inversely proportional to NC with a high feed rate (0.2 mm/rev) (Figure 5b). The explanation is that, for high feed rate, the cross section of the cutting layer is large, which means more space to create favorable conditions for the formation of $MoS_2$ oil mist, even with a high concentration of $MoS_2$ nanoparticles up to 3.0 wt%. The impedance phenomenon still does not occur, and the good lubricating condition effectively reduces surface roughness.

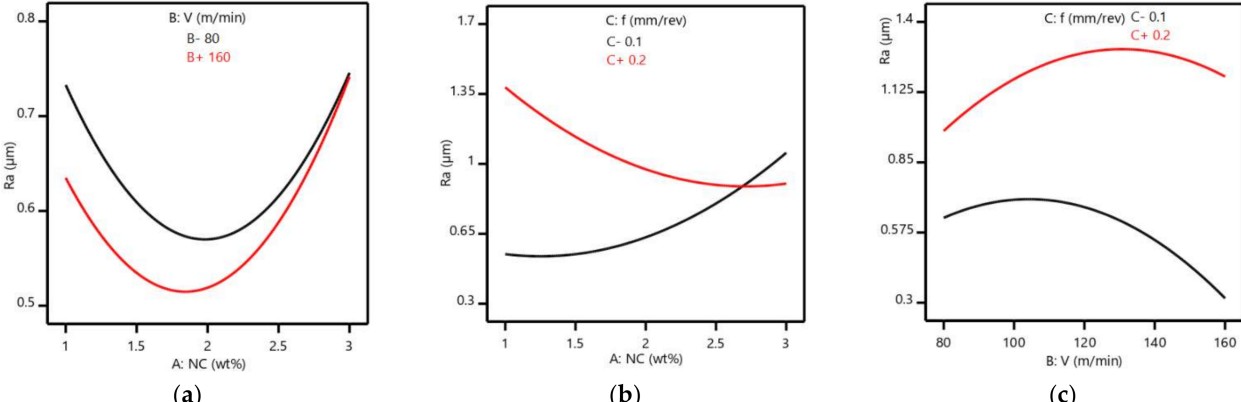

**Figure 5.** Interaction effects between survey parameters on surface roughness: (**a**) cutting speed and nanoparticle concentration; (**b**) feed rate and nanoparticle concentration; (**c**) feed rate and cutting speed.

The interaction between feed rate and cutting speed had a strong effect on surface roughness as shown in Figure 5c. The surface roughness reached a small value with high cutting speed and low feed rate. Thus, a small cutting speed combined with high nanoparticle concentration is recommended in rough cutting with a large feed rate. For finishing, a high cutting speed, low feed rate, and low nanoparticle concentration are suggested.

The influence between the cutting speed and the concentration of $MoS_2$ nanoparticles on the surface roughness value corresponding to the different feed rates is shown in Figures 6–8. The surface plot shows that the surface roughness reached a smaller value with a high cutting speed and a medium level of NC. The contour plot shows the effect trend and exhibits the optimal domain corresponding to the appropriate technological parameters. Figure 6 shows that, at f = 0.1 mm/rev, the surface roughness values decreased with the increasing cutting speed combined with the reduction of nanoparticle concentration. In finishing machining (f = 0.1 mm/rev), the range of surface roughness was less than 0.3 μm when the cutting speed was greater than 150 m/min, and the nanoparticle concentration was about 1.0 wt%. Figure 7 indicates that, at f = 0.15 mm/rev, the surface roughness values tended to decrease with the nanoparticle concentration at the medium level (2.0 wt%) in combination with either a high cutting speed of 160 m/min or a low cutting speed of 80 m/min. The surface roughness was less than 0.6 μm when the cutting speed was more than 150 m/min and the nanoparticle concentration was 2.0 wt%. The results in Figure 8 show that, at f = 0.2 mm/rev, the surface roughness reached the smallest values with the decreasing cutting speed combined with the growing nanoparticle concentration. For rough cutting (f = 0.2 mm/rev), the surface roughness was less than 0.8 μm when the cutting speed was less than 90 m/min and the nanoparticle concentration was about 3.0 wt%. It can be observed that there is an interaction effect between the feed rate and nanoparticle concentration, which strongly influences the surface roughness. When increasing the feed rate, the nanoparticle concentration should be raised to maintain the good lubricating performance and surface quality. The main reason is that the cross section of the cut layer and the dynamic clearance angle rise with the increasing feed rate, so more space is needed to be filled up with the tribofilm. Therefore, a higher nanoparticle concentration is more favorable. For improving the productivity, the cutting speed V = 160 m/min can be selected with the low feed rate of 0.1 mm/rev and low nanoparticle concentration of 1.0 wt% to achieve the smallest surface roughness values (Figure 9). However, optimization must be carried out to find the exact optimal values.

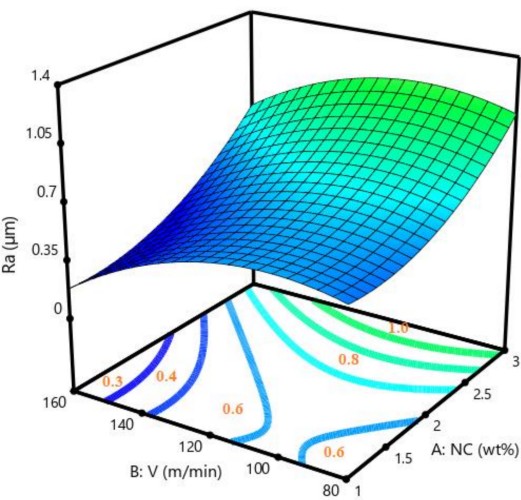

**Figure 6.** Effect of cutting speed and nanoparticle concentration on surface roughness with f = 0.1 mm/rev.

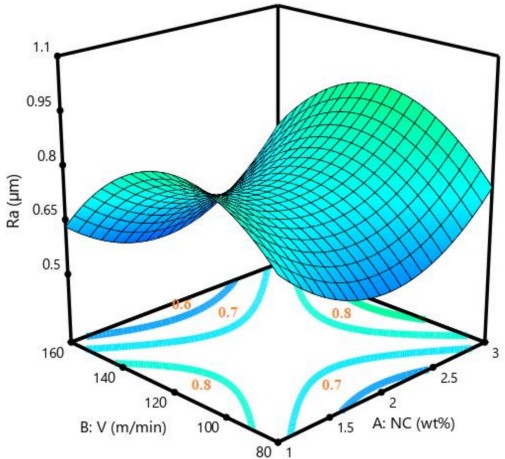

**Figure 7.** Effect of cutting speed and nanoparticle concentration on surface roughness with f = 0.15 mm/rev.

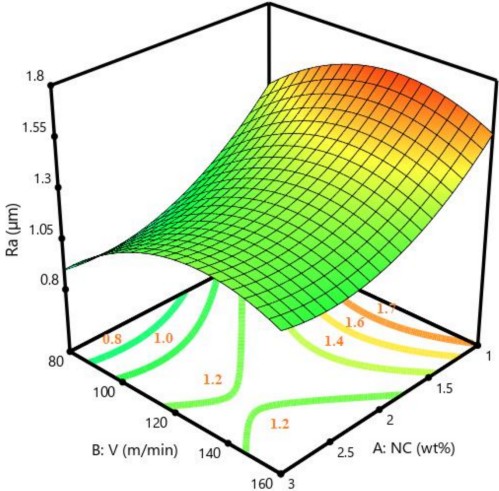

**Figure 8.** Effect of cutting speed and nanoparticle concentration on surface roughness with f = 0.2 mm/rev.

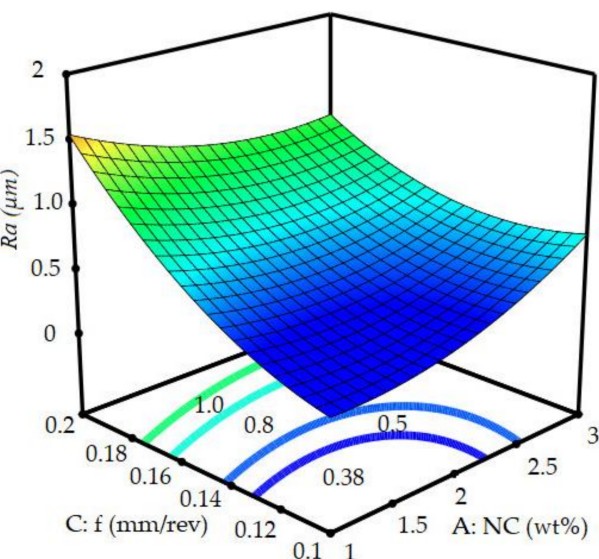

**Figure 9.** Effect of feed rate and nanoparticle concentration on surface roughness with V = 160 m/min.

### 3.3. Surface Roughness Optimization

The optimal cutting parameters for surface roughness in hard turning of 90CrSi steel under the MoS$_2$ NFMQL condition were determined through the optimization module on Design-Expert software 11. The selected minimization objective is because smaller surface roughness is better. The optimization results are shown in Figure 10. The surface roughness value reached the minimum predicted value at 0.21 µm with a MoS$_2$ nanoparticle concentration of 1.2 wt%, cutting speed of 160 m/min, and feed rate of 0.1 mm/rev.

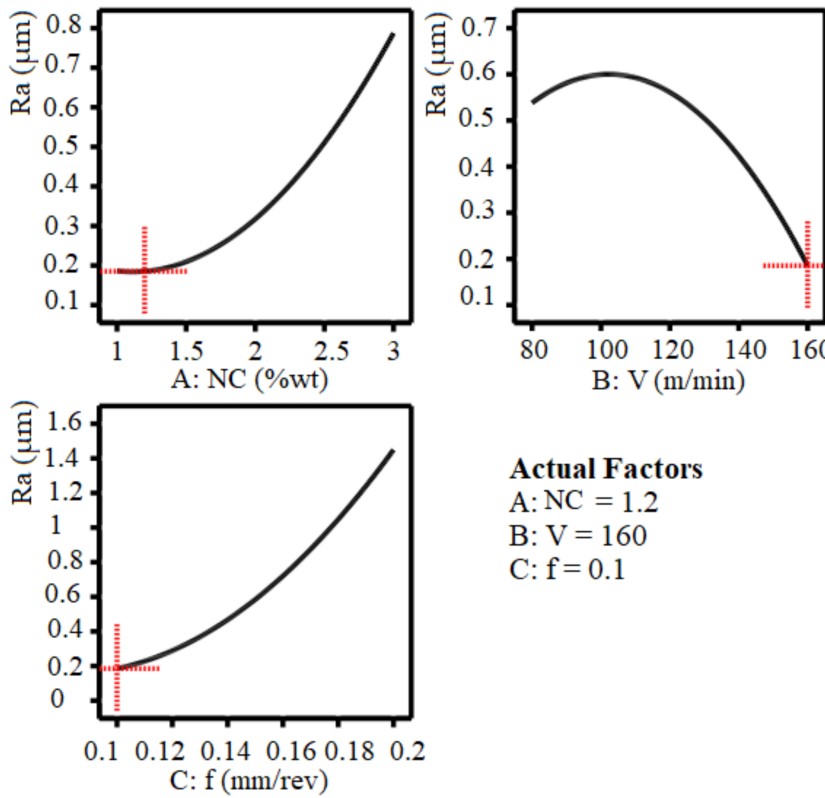

**Figure 10.** Optimization results of surface roughness.

### 3.4. Investigation of Surface Microstructure and Topography

Based on the optimization results of surface roughness, the experimental test for validation was carried out. The obtained surface roughness was 0.232 μm, which is a 10% difference from the predicted value from the optimization results, which is acceptable [35] because the actual hard machining process is subject to some other influencing factors, especially vibration. The microstructure and topography of the machined surface under pure MQL and $MoS_2$ NFMQL conditions were studied by using a KEYENCE VHX-7000 Digital Microscope, and the obtained images are shown in Figures 11 and 12. It can be clearly seen in Figure 11a that the white layer on the machined surface microstructure was caused by the high cutting temperature, because the MQL technique can provide good lubrication but the cooling effect is low. The heat deterioration causes the deformation reflected by the surface topography (Figure 11b). The surface deformation caused by cutting temperature was much reduced under the $MoS_2$ NFMQL condition (Figure 12), which proves that $MoS_2$ nanoparticles contribute to improving the cooling and lubricating of the based oil as well as the surface quality.

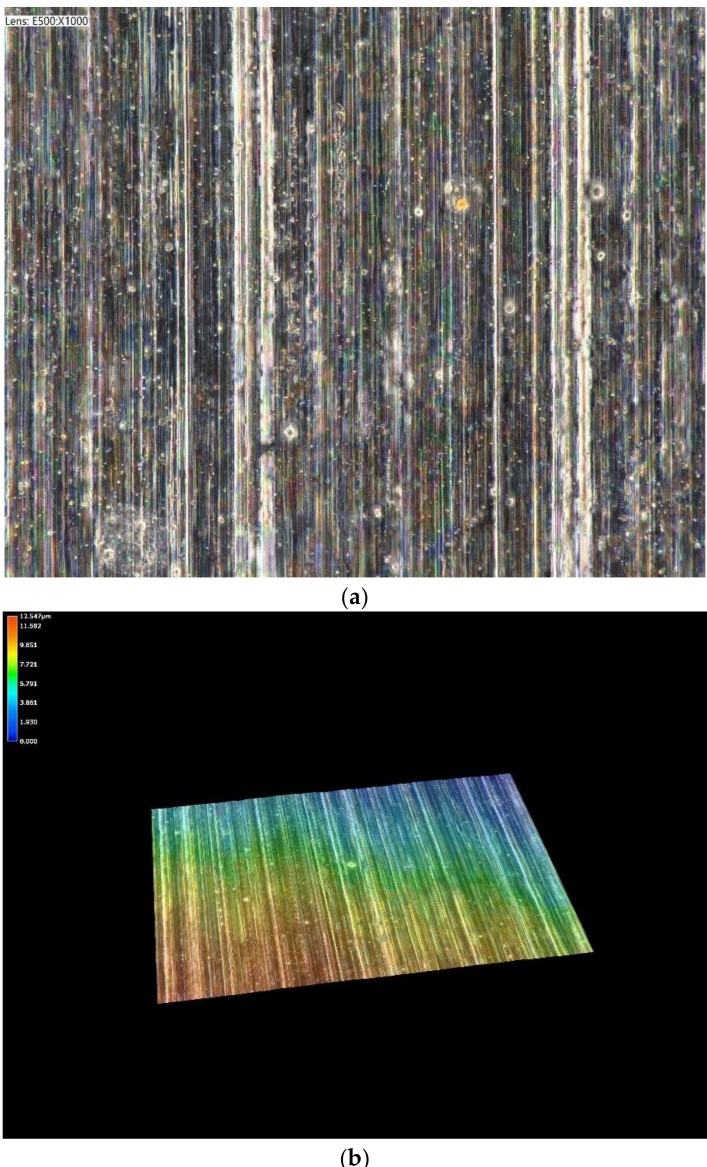

(**a**)

(**b**)

**Figure 11.** (**a**) Surface microstructure and (**b**) surface topography under pure MQL condition.

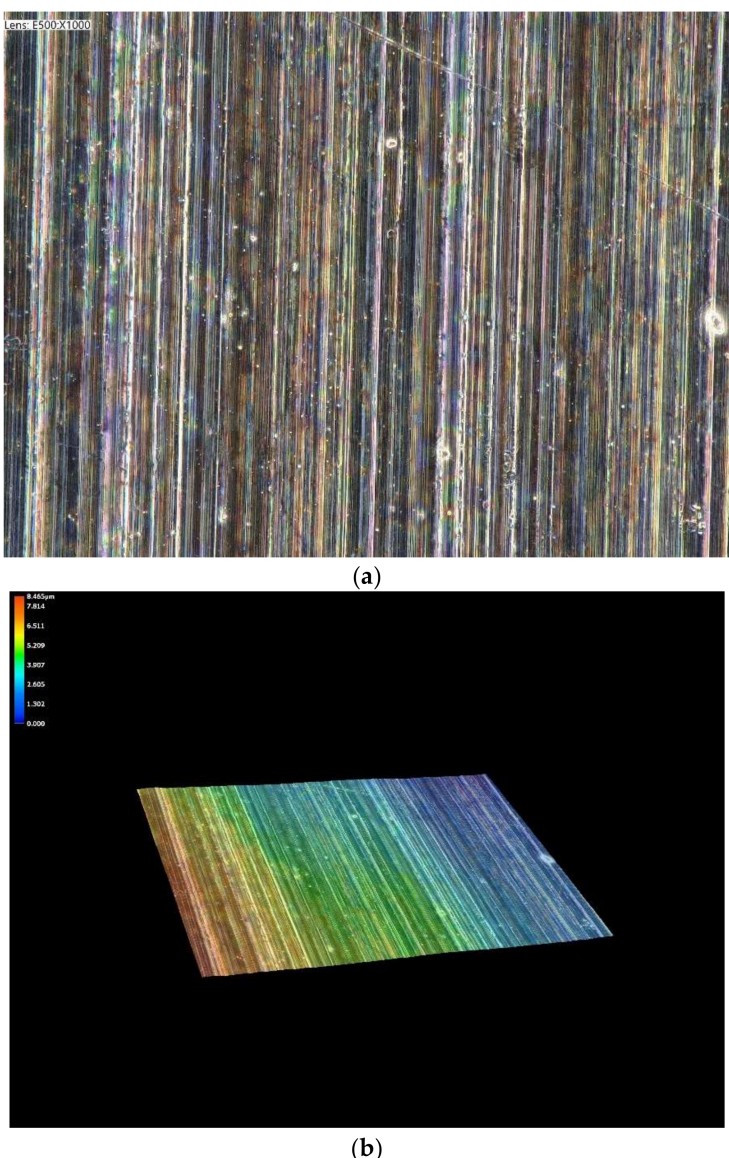

**Figure 12.** (**a**) Surface microstructure and (**b**) surface topography under MoS$_2$ NFMQL condition.

## 4. Conclusions

A Box–Behnken experimental design and ANOVA analysis were utilized to evaluate the effects of variables on the objective functions. The influence and interaction effects of nanoparticle concentration, cutting speed, and feed rate on surface roughness in MoS$_2$ NFMQL hard turning process were studied

The surface roughness, surface microstructure, and surface topography of the machined surface were better under the MQL using MoS$_2$ nano additives in an oil-in-water emulsion when compared to pure MQL. The enhancement of the lubricating characteristic created from MoS$_2$ tribofilm formation was observed. The white layers, burn marks, and surface deformation were much reduced by using the MoS$_2$ NFMQL condition due to the better cooling and lubricating performance.

The application of MQL using nano cutting oil in hard machining brings out a promising alternative solution for dry and flood conditions, improves machinability of carbide tools for cutting hard materials, and contributes to reducing the manufacturing cost as well as countering the climate change issue.

The MoS$_2$ nanoparticle concentration in oil-in-water emulsion for MQL hard turning was investigated and optimized by experiments. The optimized value was 1.2 wt% with a cutting speed of 160 m/min and a feed rate of 0.1 mm/rev.

In further work, more investigation should be focused on the $MoS_2$ nanoparticle size, lubricating mechanism, surface texture, and white layer. In addition, more focus will be given to investigate the application of hybrid nano cutting fluid containing $MoS_2$ nanoparticles.

**Author Contributions:** Conceptualization, N.M.T. and T.T.L.; Data curation, T.B.N. and T.L.T.; Formal analysis, N.M.T., T.B.N. and T.T.L.; Funding acquisition, N.M.T. and T.T.L.; Investigation, N.M.T. and T.L.T.; Methodology, T.T.L.; Project administration, N.M.T.; Resources, T.B.N. and T.L.T.; Software, T.B.N. and T.L.T.; Supervision, N.M.T. and T.T.L.; Validation, N.M.T., T.B.N. and T.T.L.; Visualization, N.M.T. and T.T.L.; Writing—original draft, N.M.T. and T.T.L.; Writing—review & editing, N.M.T. and T.T.L. All authors have read and agreed to the published version of the manuscript.

**Funding:** This research was funded by Thai Nguyen University of Technology, Thai Nguyen University, Vietnam.

**Acknowledgments:** The work presented in this paper is supported by Thai Nguyen University of Technology, Thai Nguyen University, Vietnam.

**Conflicts of Interest:** The authors declare no conflict of interest.

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
