# Peer review of "Investigation of the Effects of Nanoparticle Concentration and Cutting Parameters on Surface Roughness in MQL Hard Turning Using MoS2 Nanofluid"

_fluids, doi:10.3390/fluids6110398_

Round 1
Reviewer 1 Report
- Language must be improved in the manuscript.
- Figures 4, 5, and 6: The author needs to include the observations and discussions regarding the contour plots.
- In figure 5: Justify the reason for which the contour plot of surface finish at feed rate f=0.15 mm/rev is different from the other feed rates.
- From figure 2a the minimum surface finish was observed at 2% of concentration. In the 25th run of table 2, the surface roughness obtained is 0.381 µm at conditions, the concentration of 2 wt%, cutting speed of 160 m/min, and feed rate of 0.1 mm/rev. The optimum conditions obtained (Section 3. c) are the concentration of 1.2 wt%, cutting speed of 160 m/min, and feed rate of 0.1 mm/rev and obtained 21 µm surface roughness. According to the above 2 statements, the surface roughness should be more than 0.381 µm. how does the author justify the optimized result?
- The conclusion given by the author is not accurate and precise and it is advised to modify.
Author Response
RESPONSES TO THE REVIEWER 1
We are very grateful for the reviews provided by the editors and each of the external reviewers of this manuscript. Please see below, our detailed response to comments.
- Language must be improved in the manuscript.
Answer:
Language was revised carefully by following the reviewer’s comments
- Figures 4, 5, and 6: The author needs to include the observations and discussions regarding the contour plots.
Answer:
The discussions regarding the contour plots were extended in the revised manuscript. The changes are in red color
In figure 5: Justify the reason for which the contour plot of surface finish at feed rate f=0.15 mm/rev is different from the other feed rates.
Answer:
The contour plot of surface finish shows the interaction of different feed rates and nanoparticle concentrations was added in the manuscript
- From figure 2a the minimum surface finish was observed at 2% of concentration. In the 25th run of table 2, the surface roughness obtained is 0.381 µm at conditions, the concentration of 2 wt%, cutting speed of 160 m/min, and feed rate of 0.1 mm/rev. The optimum conditions obtained (Section 3. c) are the concentration of 1.2 wt%, cutting speed of 160 m/min, and feed rate of 0.1 mm/rev and obtained 0.21 µm surface roughness. According to the above 2 statements, the surface roughness should be more than 0.381 µm. how does the author justify the optimized result?
Answer:
Figure 2a shows the effects of investigated variables on the average roughness value, and Figure 7 shows the optimal value predicted by the mathematical model. The validation experiment was conducted and the surface roughness and microstructure in Figure …were added and expanded the discussion in the revised manuscript.
- The conclusion given by the author is not accurate and precise and it is advised to modify.
Answer:
The conclusion was re-written by following the reviewer’s comments. The changes are in red color.

Reviewer 2 Report
- In this article, I haven't seen any photos of real experimental equipment and schematic diagrams of the display equipment. The article should contain photos of real experimental equipment so that readers can more easily understand the process of the experiment and the conduct of the experiment.
- The author used the 90CrSi steel in this experiment. The author should explain its material properties. For example: Why choose 90CrSi steel for experiments and information on the physical properties of 90CrSi steel.
-
Line 88 of this paragraph
“However, most of the studies only focused on the efficiency of the machining process using MoS2 nano cutting fluid compared to other conditions without using nanoparticles, and the provison of the set of machining parameters and their optimized values is still limited, especially for hard turning process.”
At present, many related studies can be compared; and you can refer to the following articles.
IEEE TRANSACTIONS ON AUTOMATION SCIENCE AND ENGINEERING, VOL. 15, NO. 3, JULY 2018
DOI:10.1109/TASE.2017.2726000
IEEE TRANSACTIONS ON INDUSTRIAL INFORMATICS, VOL. 16, NO. 8, AUGUST 2020
DOI: 10.1109/TII.2019.2955736 -
In this article, the author mentioned "MoS2 nanoparticles manufactured by China Luoyang have the layered structure with the average size of 30 nm."
Why did the author use this size of MoS2 nanoparticles? If used in other sizes, what will be the impact? Different nanoparticle sizes also have a certain level of surface roughness degree of influence
You can refer to and compare this article.
Int. J. Fuzzy Syst. (2020) 22(7):2101–2118
https://doi.org/10.1007/s40815-020-00930-w -
This article adopts the Box-Behnken design experimental method, but I did not see the further introduction of this research method in the article and the literature review. Is the suitability of this method used in this study? What is the difference between Box-Behnken design and other methods such as Taguchi Method and Uniform Design?
-
This article has established a regression model to predict surface roughness. However, the tested and calculated data are all known experimental data. How can it be explained that this regression model can accurately predict unknown real experimental values?
-
The following three paragraphs in the article are analyzed and compared,
- Independent influence of the input variables on surface roughness Ra
- Interaction effects between input machining variables on the surface roughness
- Surface roughness optimization
Author Response
RESPONSES TO THE REVIEWER 2
We are very grateful for the reviews provided by the editors and each of the external reviewers of this manuscript. Please see below, our detailed response to comments.
- In this article, I haven't seen any photos of real experimental equipment and schematic diagrams of the display equipment. The article should contain photos of real experimental equipment so that readers can more easily understand the process of the experiment and the conduct of the experiment.
Answer:
Thank you very much. The real experimental equipment was added in the revised manuscript
- The author used the 90CrSi steel in this experiment. The author should explain its material properties. For example: Why choose 90CrSi steel for experiments and information on the physical properties of 90CrSi steel.
Answer:
The table of chemical composition of 90CrSi steel was added and shown in Table 1. The physical and mechanical properties of 90CrSi steel. The reason for choosing 90CrSi steel for experiments was added in the revised manuscripts by following the reviewer’s comments.
- Line 88 of this paragraph
“However, most of the studies only focused on the efficiency of the machining process using MoS2 nano cutting fluid compared to other conditions without using nanoparticles, and the provison of the set of machining parameters and their optimized values is still limited, especially for hard turning process.”
At present, many related studies can be compared; and you can refer to the following articles.
IEEE TRANSACTIONS ON AUTOMATION SCIENCE AND ENGINEERING, VOL. 15, NO. 3, JULY 2018
DOI:10.1109/TASE.2017.2726000
IEEE TRANSACTIONS ON INDUSTRIAL INFORMATICS, VOL. 16, NO. 8, AUGUST 2020
DOI: 10.1109/TII.2019.2955736
Answer:
The suggested articles were revised carefully and added in the Reference section. Line 88 was rewritten and modified by following the reviewer’s comments.
- In this article, the author mentioned "MoS2 nanoparticles manufactured by China Luoyang have the layered structure with the average size of 30 nm."
Why did the author use this size of MoS2 nanoparticles? If used in other sizes, what will be the impact? Different nanoparticle sizes also have a certain level of surface roughness degree of influence
You can refer to and compare this article.
Int. J. Fuzzy Syst. (2020) 22(7):2101–2118
https://doi.org/10.1007/s40815-020-00930-w
Answer:
Thank you very much for very helpful comments. The grain size of MoS2 nanoparticles has a certain influence level on surface roughness. The smaller size is more favorable for finishing. The suggested article was referred and added to provide the base for choosing the nanoparticle size. Thank you very much.
- This article adopts the Box-Behnken design experimental method, but I did not see the further introduction of this research method in the article and the literature review. Is the suitability of this method used in this study? What is the difference between Box-Behnken design and other methods such as Taguchi Method and Uniform Design?
Answer:
Thank you very much. In this article, the authors want to find the optimal condition, so we use this method. Taguchi and Uniform have the advantage of surveying many influencing factors at the same time and show the influence trend. However, for predicting the optimal value, the Box-Behnken method has more favorable. The advantages and reasons for choosing Box-Behnken design were added in the revised manuscript.
- This article has established a regression model to predict surface roughness. However, the tested and calculated data are all known experimental data. How can it be explained that this regression model can accurately predict unknown real experimental values?
Answer:
Thank you very much. This is a new research topic, so there are still many issues that are needed to be investigated and studied further for each specific cutting condition. In the survey range, this regression model can be used to predict the surface roughness value; however, further studies are needed. The authors totally agree the reviewer’s comments and continue to make further investigation.
- The following three paragraphs in the article are analyzed and compared,
- Independent influence of the input variables on surface roughness Ra
- Interaction effects between input machining variables on the surface roughness
- Surface roughness optimization
The author is requested to add pictures of tool wear or workpiece surface conditions in this section to increase the article's readability.
Answer:
Thank you very much for very helpful comments. The images of surface microstructure and topography were added to make further investigation and validation for the optimization results.

Reviewer 3 Report
Below I am sending the main observations about the article titled "Investigation of the effects of nanoparticle concentration and cutting parameters on surface roughness in MQL hard turning using MoS2 nanofluid".
Why did the authors decide to use a carbide tool (according to the manufacturer, the T9125 grade is not intended for turning hardened steel) and not, for example, from CBN or ceramics?
The assessment of the surface quality by only the Ra parameter is incomplete. It would be worth supplementing the analysis with other amplitude parameters.
The conclusions are rather obvious without deeper scientific contribution.
Author Response
RESPONSES TO THE REVIEWER 3
We are very grateful for the reviews provided by the editors and each of the external reviewers of this manuscript. Please see below, our detailed response to comments.
Below I am sending the main observations about the article titled "Investigation of the effects of nanoparticle concentration and cutting parameters on surface roughness in MQL hard turning using MoS2 nanofluid".
- Why did the authors decide to use a carbide tool (according to the manufacturer, the T9125 grade is not intended for turning hardened steel) and not, for example, from CBN or ceramics?
Answer:
Thank you very much for very helpful comments. The authors aimed to use normal carbide inserts because they are commonly used in industry with low cost and high toughness. By applying NFMQL technology, it is still possible to improve the machinability of carbide inserts and use effectively for hard turning, which contributes to reducing machining costs.
- The assessment of the surface quality by only the Ra parameter is incomplete. It would be worth supplementing the analysis with other amplitude parameters.
Answer:
Thank you very much for very helpful comments. The images of machined surface microstructure and topography were added to make further investigation and validation for the optimization results.
The conclusions are rather obvious without deeper scientific contribution.
Answer:
The conclusion was rewritten by following the reviewer’s comments

Reviewer 4 Report
1) The novelty of the conducted research is not very strong.
2) Compared with the existing research, what are the main research contributions of the present research?
3) In this article, it is necessary to discuss in detail the lubrication mechanism of MoS2 nanofluid for reducing the machined surface roughness compared with pure MQL.
4) There are too many keywords.
5) In introduction section, the literature review is too general and some relevant papers dealing with the investigation of the effects of nanoparticle concentration and cutting parameters on surface roughness in MQL hard turning using MoS2 nanofluid are missing.
Moreover, the deeper discussion of the obtained literature is necessary. Therefore, the authors should revise it and state clearly why their work is relevant, and which the expected advances in knowledge are.
6) It is necessary to make more and detailed comments on the working principle of the experimental set up in Fig. 1.
7) It is necessary to clearly indicate the distance between the MQL nozzle and the machining zone in this article.
8) It is necessary to add the sources of the workpiece material and specific dimension of the length the workpiece material.
9) It is necessary to provide the more technical details of MoS2 nanoparticles.
10) In this article, it is necessary to add the experimental details of the dispersion and stability of the prepared nanofluids. What method is used to judge the dispersion and stability of the prepared nanofluids? How is the dispersion and stability of the prepared nanofluids?
11) In this article, it is necessary to clearly indicate the specific references and justifications for choosing MQL experimental parameters (air pressure, air flow rate).
12) In this article, it is necessary to clearly point out the more experimental details for measuring surface roughness during turning process.
13) Why choose Box-Behnken experimental design?
14) In Table 1, it is necessary to clearly indicate the specific references and justifications for choosing turning parameters and their levels (nanoparticle concentration, cutting speed, feed rate).
15) The author wrote that “the coefficient of determination R2= 87.57%”. But, it seems that the value of R2 is not very high. Hence, it is necessary to provide the reasonable and adequate explains for the value of R2.
16) In Fig. 2, it is necessary to more reasonably and fully explain why the surface roughness increases with the increase of nanoparticle concentration.
17) “As the cutting speed increases from 80 to 120 m/min, the surface roughness grows up but decreases when cutting speed rises from 120 to 160 m/min.”
It is necessary to provide the more reasonable and adequate explanations for the above research findings.
18) Why does the surface roughness increase with the increase of feed rate?
19) “The surface roughness value is proportional to NC with low feed rate (0.1 mm/rev) but inversely proportional to NC with high feed rate (0.2 mm/rev) (Figure 3b).”
Similarly, it is necessary to provide the more reasonable and adequate explanations for the above research findings.
20) In this article, it is necessary to make more comments on the variation trends of surface roughness with cutting speed, nanoparticle concentration and feed rate in Fig. 4, Fig. 5 and Fig. 6. Furthermore, it necessary to provide the more reasonable and sufficient explanations for the above changing trends of surface roughness.
21) In this article, it is necessary to add more details regarding the surface roughness optimization.
22) The abstract and conclusion section need to be improved.
23) It is found that the manuscript is incomplete in all sense and the main discussion section is minimum.
24) The results are mainly presented by figures and tables. It is more important to provide the sufficient and reasonable explanation for all the research results. This is also the weakest aspect of the study.
25) Results and discussion should be modified accordingly to the aim of the article.
26) The format of all the tables and references should be modified according to the journal guidelines.
Author Response
RESPONSES TO THE REVIEWER 4
We are very grateful for the reviews provided by the editors and each of the external reviewers of this manuscript. Please see below, our detailed response to comments.
1) The novelty of the conducted research is not very strong.
Answer:
The manuscript was revised entirely and modified carefully to improve the quality.
2) Compared with the existing research, what are the main research contributions of the present research?
Answer:
The manuscript was revised and the main research contributions were added
3) In this article, it is necessary to discuss in detail the lubrication mechanism of MoS2 nanofluid for reducing the machined surface roughness compared with pure MQL.
Answer:
Thank you very much for very helpful comments. The manuscript was revised to expand the lubrication mechanism of MoS2 nanofluid through the surface microstructure and topography, which were compared with pure MQL. The discussion was expanded by following the reviewer’s comments.
4) There are too many keywords.
Answer:
Some keywords were removed by following the reviewer’s comments.
5) In introduction section, the literature review is too general and some relevant papers dealing with the investigation of the effects of nanoparticle concentration and cutting parameters on surface roughness in MQL hard turning using MoS2 nanofluid are missing.
Moreover, the deeper discussion of the obtained literature is necessary. Therefore, the authors should revise it and state clearly why their work is relevant, and which the expected advances in knowledge are.
Answer:
The literature review was revised carefully and modified to show the deeper discussion of the obtained work. Some relevant papers dealing with the investigation of the effects of nanoparticle concentration and cutting parameters on surface roughness in MQL hard turning using MoS2 nanofluid were added by following the reviewer’s comments. The changes are in red color.
6) It is necessary to make more and detailed comments on the working principle of the experimental set up in Fig. 1.
Answer:
Thank you very much. The real experimental equipment was added in the revised manuscript
7) It is necessary to clearly indicate the distance between the MQL nozzle and the machining zone in this article.
Answer:
The MQL nozzle distance and spray angle were added in the revised manuscript, which is based on the previous study. Thank you very much.
8) It is necessary to add the sources of the workpiece material and specific dimension of the length the workpiece material.
Answer:
Thank you very much. The table of chemical composition of 90CrSi steel was added and shown in Table 1. The physical and mechanical properties of 90CrSi steel. The length of workpiece was added. The reason for choosing 90CrSi steel for experiments was added in the revised manuscripts by following the reviewer’s comments.
9) It is necessary to provide the more technical details of MoS2 nanoparticles.
Answer:
The TEM image and the purity of MoS2 nanoparticles were added in the revised manuscripts
10) In this article, it is necessary to add the experimental details of the dispersion and stability of the prepared nanofluids. What method is used to judge the dispersion and stability of the prepared nanofluids? How is the dispersion and stability of the prepared nanofluids?
Answer:
Thank you very much. 3000868-Ultrasons-HD was used for uniform dispersion of nanoparticles in the based oil. In order to use the obtained nanofluids effectively and avoid the precipitation of agglomerated nanoparticles during the long time of machining, the nanofluids were placed in the 3000868-Ultrasons-HD as shown in the following image and directly used for MQL system.
11) In this article, it is necessary to clearly indicate the specific references and justifications for choosing MQL experimental parameters (air pressure, air flow rate).
Answer:
Thank you very much. The base for choosing MQL experimental parameters (air pressure, air flow rate) was added and cited in the revised manuscript.
12) In this article, it is necessary to clearly point out the more experimental details for measuring surface roughness during turning process.
Answer:
Thank you very much. The surface roughness Ra was measured by 3 times and taken by the average value after each trial by Mitutoyo SJ210. The image for measuring surface roughness for turning process is shown as below.
13) Why choose Box-Behnken experimental design?
Answer:
Thank you very much. In this article, the authors want to find the optimal condition, so we use this method. Taguchi and Uniform have the advantage of surveying many influencing factors at the same time and show the influence trend. However, for predicting the optimal value, the Box-Behnken method has more favorable. The advantages and reasons for choosing Box-Behnken design were added in the revised manuscript.
14) In Table 1, it is necessary to clearly indicate the specific references and justifications for choosing turning parameters and their levels (nanoparticle concentration, cutting speed, feed rate).
Answer:
Thank you very much. The base for choosing turning parameters and their levels (nanoparticle concentration, cutting speed, feed rate) was added and cited in the revised manuscript.
15) The author wrote that “the coefficient of determination R2= 87.57%”. But, it seems that the value of R2 is not very high. Hence, it is necessary to provide the reasonable and adequate explains for the value of R2.
Answer:
Thank you very much. The coefficient of determination R2 is one of the parameters to evaluate the fit, and R2 greater than 50% can be considered using the model to evaluate the influence of the parameters on the objective function [33]
16) In Fig. 2, it is necessary to more reasonably and fully explain why the surface roughness increases with the increase of nanoparticle concentration.
Answer:
Thank you very much for very helpful comments. The increase in MoS2 nanoparticle concentration from 1.0% to 2.0% helps to reduce the surface roughness, but the surface roughness value increases when rising the concentration from 2.0% to 3.0%, which agrees with the obtained results of Zhang Dongkun et al. [18]. There is an interaction effect of nanoparticle type and based cutting oil, so there exists an appropriate value of concentration for each type of nano cutting fluid to promote the best cooling and lubricating effect. For MoS2 nanoparticles, the formation of tribo film is very sensitive with the nano concentration value, so the increase in MoS2 nanoparticle concentration from 2.0% to 3.0% makes the distance among particles become smaller, and collisions among nanoparticles increase. Hence, the formation of tribo film is unfavorable, which causes the negative effect on the lubricating performance and deteriorate the surface roughness.
17) “As the cutting speed increases from 80 to 120 m/min, the surface roughness grows up but decreases when cutting speed rises from 120 to 160 m/min.”
It is necessary to provide the more reasonable and adequate explanations for the above research findings.
Answer:
Thank you very much for very helpful comments. As the cutting speed increases from 80 to 120 m/min, the surface roughness grows up but decreases when cutting speed rises from 120 to 160 m/min, and the reasons were previously discussed by [31,32]. On the one hand, at low cutting speed, the heat generation required to reduce the shear strength of workpiece material is not enough, so higher cutting forces will be required, affecting the surface finish negatively. In addition, the nanoparticles easily adhere to the top of the cutting edge, causing scratches on the machined surface. This phenomenon is formed and lost continuously, thus deteriorating surface quality. For high cutting speed, the workpiece shear strength decreases because the cutting heat is high enough to reduce the required cutting forces and positively affect the machined surface finish. On the other hand, the amount of nano cutting oil penetrates in cutting zone without being impeded and does not adhere to the top of the cutting edge, so the surface roughness value decreases.
18) Why does the surface roughness increase with the increase of feed rate?
Answer:
Thank you very much. Because in hard machining, the formation of surface roughness is mainly due to the geometric cause, which is the scratches of cutting tool on machined surface [1,28], and the influence of other factors is not significant.
19) “The surface roughness value is proportional to NC with low feed rate (0.1 mm/rev) but inversely proportional to NC with high feed rate (0.2 mm/rev) (Figure 3b).”
Similarly, it is necessary to provide the more reasonable and adequate explanations for the above research findings.
Answer:
Thank you very much for very helpful comments. When increasing the feed rate, the nanoparticle concentration should be risen to maintain the good lubricating performance and surface quality. The main reason is that the cross section of the cut layer and the dynamic clearance angle rise with the increasing feed rate, so more space is needed to be filled up with the tribo film. Therefore, higher nanoparticle concentration is more favorable. The surface plot was added to show the relation between the feed rate and nanoparticle concentration, and more explanations was discussed for the findings in the revised manuscript.
20) In this article, it is necessary to make more comments on the variation trends of surface roughness with cutting speed, nanoparticle concentration and feed rate in Fig. 4, Fig. 5 and Fig. 6. Furthermore, it necessary to provide the more reasonable and sufficient explanations for the above changing trends of surface roughness.
Answer:
Thank you very much. The discussion on the variation trends of surface roughness with cutting speed, nanoparticle concentration and feed rate in Fig. 4, Fig. 5 and Fig. 6 was added to provide the more reasonable and sufficient explanations.
21) In this article, it is necessary to add more details regarding the surface roughness optimization.
Answer:
Thank you very much for very helpful comments. The images of surface microstructure and topography were added to make further investigation and validation for the optimization results.
22) The abstract and conclusion section need to be improved.
Answer:
The abstract and conclusion were revised and modified by following the reviewer’s comments. The changes are in red color.
23) It is found that the manuscript is incomplete in all sense and the main discussion section is minimum.
Answer:
Thank you very much for very helpful comments. The manuscript was revised and extended with the deeper discussion and experimental results to prove and validate the obtained results.
24) The results are mainly presented by figures and tables. It is more important to provide the sufficient and reasonable explanation for all the research results. This is also the weakest aspect of the study.
Answer:
Thank you very much for very helpful comments. The images of machined surface microstructure and topography were added to make further investigation on the lubricating mechanism of MoS2 nanoparticles, and the obtained results were compared with pure MQL condition. Also, the validation experiment test for the optimization results was carried out.
25) Results and discussion should be modified accordingly to the aim of the article.
Answer:
Results and discussion were revised and modified by following the reviewer’s comments.
26) The format of all the tables and references should be modified according to the journal guidelines.
Answer:
The format of the tables and references were modified according to the journal guidelines. Thank you very much.

Round 2
Reviewer 1 Report
Accepted
Author Response
RESPONSES TO THE REVIEWER 1
We are very grateful for the reviews provided by the reviewer of this manuscript.
- English language and style are fine/minor spell check required
Answer:
English language and style were revised carefully and the changes are in red color

Reviewer 2 Report
The author's supplementary correction is very complete, and I have no comments.
Author Response
RESPONSES TO THE REVIEWER 2
We are very grateful for the reviews provided by the reviewer of this manuscript.
- English language and style are fine/minor spell check required
Answer:
English language and style were revised carefully and the changes are in red color

Reviewer 4 Report
The authors have considered some comments of reviewers. Nevertheless, some aspects of the comments were ignored. The explanation of some research results is not very reasonable and sufficient. To improve the quality of the paper, the following issues need to be addressed.
1) It is necessary to make more and detailed comments on the working principle of the experimental diagram in Fig. 2.
2) It is necessary to add the sources of the workpiece material.
3) In this article, it is necessary to add the following experimental details: what method is used to judge the dispersion and stability of the prepared nanofluids? How is the dispersion and stability of the prepared nanofluids?
4) In this article, it is necessary to provide the reasonable and adequate explains for the surface roughness increase with the increase of feed rate.
5) In Fig. 11, how to prove that the features existing on the machined surface are the white layer and burn mark?
6) “Based on the optimization results of surface roughness, the experimental test for validation was carried out. The obtained surface roughness was 0.232 µm, which is 10% difference from the predicted value from the optimization result.”
Therefore, it is necessary to reasonably and adequately explain the reasons for the error between the experimental roughness value and the predicted roughness value.
Author Response
RESPONSES TO THE REVIEWER 4
We are very grateful for the reviews provided by the reviewer of this manuscript. Please see below, our detailed response to comments.
The authors have considered some comments of reviewers. Nevertheless, some aspects of the comments were ignored. The explanation of some research results is not very reasonable and sufficient. To improve the quality of the paper, the following issues need to be addressed.
1) It is necessary to make more and detailed comments on the working principle of the experimental diagram in Fig. 2.
Answer:
This is the author's shortcoming. Thank you very much for your valuable comments. The working principle of the experimental diagram in Fig. 2 was added in the revised manuscript.
2) It is necessary to add the sources of the workpiece material.
Answer:
The sources of the workpiece material was added in the revised manuscript.
3) In this article, it is necessary to add the following experimental details: what method is used to judge the dispersion and stability of the prepared nanofluids? How is the dispersion and stability of the prepared nanofluids?
Answer:
In this article, the nanoparticle is directly mixed with the base oil and the ultrasonication with ultrasonic frequency 40 kHz for 30-45 minutes [18] has been performed to ensure uniform distribution of MoS2 nanoparticles in oils as shown in following figure. Because oil-in-water emulsion has very low kinematic viscosity (about 37.108 mm2/s at 40ºC from the work of Rajaguru, J. and Arunachalam, N. (2020)), the obtained nanofluid was still put in the Ultrasons–HD Ultrasonicator and directly used for MQL system to avoid the agglomeration, which is presented in the set up diagram in Figure 2.
The dispersion and stability of the nano-cutting oil were currently tested by our experimental observations and based on the other research results, which were cited in the revised manuscript. We really appreciate the reviewer's comments and will conduct deeper research on the dispersion and stability of the nanoparticles in the base oil.
4) In this article, it is necessary to provide the reasonable and adequate explains for the surface roughness increase with the increase of feed rate.
Answer:
The effect of feed rate on surface roughness is the strongest as feed rate per tooth increases, the distances between peaks and valleys will be higher [Ref. 1, p.119]. The reasonable explanation was added to the revised manuscript. Thank you very much
5) In Fig. 11, how to prove that the features existing on the machined surface are the white layer and burn mark?
Answer:
The white layer and burn mark in this article was derived from the author's prediction based on observation. However, to confirm it, it is necessary to conduct more studies on capturing the surface layer structure on SEM microscopy. This content will be absorbed and added in the future work in the conclusion section. We really appreciate the comments of the reviewer. The content of the article has been revised.
6) “Based on the optimization results of surface roughness, the experimental test for validation was carried out. The obtained surface roughness was 0.232 µm, which is 10% difference from the predicted value from the optimization result.”
Therefore, it is necessary to reasonably and adequately explain the reasons for the error between the experimental roughness value and the predicted roughness value.
Answer:
The optimal model was built up by basing on three input factors; however, the actual machining process is subject to some other influencing factors, especially vibration, ambient temperature, so in the experimental studies, an error of 10% or lower is acceptable [35]. The reasonable explanation was added to the revised manuscript. Thank you very much for your valuable comment.

Round 3
Reviewer 4 Report
The format of all the tables and references should be modified according to the journal guidelines.